# MCH-R1 Antagonist GPS18169, a Pseudopeptide, Is a Peripheral Anti-Obesity Agent in Mice

**DOI:** 10.3390/molecules26051291

**Published:** 2021-02-27

**Authors:** Jean A. Boutin, Magali Jullian, Lukasz Frankiewicz, Mathieu Galibert, Philippe Gloanec, Thierry Le Diguarher, Philippe Dupuis, Amber Ko, Laurent Ripoll, Marc Bertrand, Anne Pecquery, Gilles Ferry, Karine Puget

**Affiliations:** 1Institut de Recherches Internationales Servier, 92284 Suresnes, France; laurent.ripoll@servier.com; 2Genepep SA, 34430 Saint Jean de Vedas, France; magali.jullian@genepep.com (M.J.); l.frankiewicz@pracowniacms.pl (L.F.); mathieu.galibert@genepep.com (M.G.); karine.puget@genepep.com (K.P.); 3Institut de Recherches Servier, 92150 Suresnes, France; phgloanec@gmail.com; 4Technologie Servier, 45520 Gidy, France; thierry.le-diguarher@servier.com (T.L.D.); m.bertrand@servier.com (M.B.); 5Eurofins Discovery, 86600 Celle l’Evescault, France; PhilippeDupuis@eurofins.com (P.D.); AnnePecquery@eurofins.com (A.P.); 6Eurofins Discovery, New Taipei City 24891, Taiwan; AmberKo@eurofins.com; 7Institut de Recherches Servier, 78290 Croissy-sur-Seine, France; gilles.ferry@servier.com

**Keywords:** diet-induced obesity, MCH, MCH-R1 antagonist, pseudopeptide, treatment

## Abstract

Melanin-concentrating hormone (MCH) is a 19 amino acid long peptide found in the brain of animals, including fishes, batrachians, and mammals. MCH is implicated in appetite and/or energy homeostasis. Antagonists at its receptor (MCH-R1) could be major tools (or ultimately drugs) to understand the mechanism of MCH action and to fight the obesity syndrome that is a worldwide societal health problem. Ever since the deorphanisation of the MCH receptor, we cloned, expressed, and characterized the receptor MCH-R1 and started a vast medicinal chemistry program aiming at the discovery of such usable compounds. In the present final work, we describe GPS18169, a pseudopeptide antagonist at the MCH-R1 receptor with an affinity in the nanomolar range and a Ki for its antagonistic effect in the 20 picomolar range. Its metabolic stability is rather ameliorated compared to its initial parent compound, the antagonist S38151. We tested it in an in vivo experiment using high diet mice. GPS18169 was found to be active in limiting the accumulation of adipose tissues and, correlatively, we observed a normalization of the insulin level in the treated animals, while no change in food or water consumption was observed.

## 1. Introduction

Obesity is a major societal and health problem, in western countries and beyond [1,2,3]. We recently saw how obesity and its complications as a co-morbidity factor [4] might threaten the health of many people in front of viral aggression [5], rendering this condition even more dangerous than “simply” those traditionally associated with overweight conditions, such as heart conditions and joint and bone fragilization, among many others. For about two decades, researchers have attempted to determine the role of several central systems in appetite control, reasoning that the smaller the food income, the smaller the weight gain, and thus the better the health condition will be. Unsuccessful attempts are too numerous to list, but one can cite neuropeptides 1 and 5, orexin, and melanin-concentrating hormone (MCH) including upstream works from our group.

One of those systems, MCH, attracted our attention because it was described as a central control of food intake and energy expenditure [6]. MCH is a rather small peptide that exists in many animals from fishes to mammals in a quite conserved sequence [7,8]. It is produced in the brain in a well-understood fashion and circulates at low concentrations in mammal blood. It mainly exerts its activity through a seven transmembrane domain receptor named MCH-R1. The molecular pharmacology of this peptide and its receptor has been explored thoroughly during the two last decades, and its implication in appetite control has been reported several times (see details in Audinot et al. [9]). For example, engineered mice with either the receptor or the hormone knockouts were lean, even under access to a Western cafeteria regimen [10,11,12]. Moreover, in hypothalamic neurons, MCH-R1 has been located in the primary cilium of the cell. In the rare Bardet Biedl syndrome, there is a defect in the assembly of the primary cilium in hypothalamic neurons and, consequently, there is ectopic activation of the MCH-R1 signaling pathway, leading to hyperphagia and a decrease in satiety responsible for obesity and being overweight in affected children [13,14]. Furthermore, outside the obesity area, MCH-R1 antagonists may help us understand the interrelation between MCH and cocaine addiction [15], Alzheimer-related memory loss [16], depression [17] and sleep [18], as well as its interplay with other central systems such as the GABAergic one [19].

High-throughput screening campaigns for antagonists at this receptor, mostly using small molecules, were reported with mixed successes [20,21,22]. For example, at the hit-to-lead stage, it was almost impossible to separate the chemical feature, leading to high affinity at MCH-R1 from those leading to a strong inhibition of the hErg channel [23,24] and its associated cardiac toxicity, a feature unacceptable at this stage of discovery/development of new anti-obesity drug candidates. Furthermore, the behavior of animals after treatment with some of those drug candidates strongly suggested that MCH was centrally involved in far more than appetite regulation or that small organic molecules (as opposed to peptides) were not specific enough of the MCH system. Those premises were recently confirmed by the evidence linking sleep and the circadian rhythm to MCH neurons as well as rapid eye movement (REM) in narcolepsy [25]. These observations led to a decrease in interest in the MCH system.

We walked a different line of research, reasoning that it was possible to find peptide analogues with interesting pharmacological characteristics. By intensively studying and modifying the hormone peptide sequence, we ended up with peptides or pseudopeptides presenting several interesting features from a peptide super agonist [26], to a radioligand, S36057 [27] and a powerful antagonist, S38151 [28,29]. S38151 had an affinity at the human MCH-R1 receptor in the low nanomolar range and was an antagonist. Furthermore, despite limited biostability in animals, as adjudged from its concentration in blood, it showed small activity in vivo in all the obesity models [29]. We anticipated that the main reason for the blood instability of the cyclic pseudopeptide S38151 was due to its fragile S-S bond. Thus, we tried to ameliorate the stability of the cyclisation by using alternative solutions to the S-S bond: a lactam bridge or a click chemistry-based triazole bridge despite its penta-atomic cycle feature in the middle of the bridge. We also noted that the number of atoms forming the peptide cycle was an important feature of the structure-activity relationship. In the present work, we describe various avenues we walked to render the pseudopeptides more stable, as well as still being able to bind to the human receptor with high affinity. Choosing among the most powerful ones, we moved to stability studies, with the intention to fully understand the compounds in comparison to previous results obtained with their more natural counterpart, S38151 [29]. Once we were convinced that GPS18169 was able to circulate without major and immediate loss, and with potentially minimal penetration into brain tissues, we tested it in the classical model of diet-induced obesity. GPS18169 successfully fulfilled the initial goal: a slight decrease in body weight with a return to almost normal insulin level.

## 2. Results

For several years, a trend to use peptides as tools [30] as well as pharmacological or therapeutic agents has risen again [31,32,33]. Despite a bad reputation, peptides tend to be better complementary tools to proteins than organic molecules could be. Some beautiful successes have been reported in the literature, as peptide drugs [34] and more and more techniques, including modifications of the peptide structures, were described (see discussion in Boutin et al. [35]).

### 2.1. Peptide Design, Syntheses, and Characterization

The main problem encountered in the MCH receptor (MCH-R1) molecular pharmacology has been the stability of the compound in in vivo situations [29]. Despite this apparent lack of stability, the potent MCH-R1 antagonist, S38151, showed interesting characteristics in all the tested obesity models [29]. The main weakness of this compound was its disulfide bridge, which potentially lacks in vivo stability. We thus concentrated our effort in exploring several alternatives to the S-S bridge by trying the three following bridge substitutes: an alkyl chain bridge, a lactam bridge and a triazole bridge. It turned out that the synthetic difficulties rendered the access to alkyl chain-substituted pseudopeptides almost impossible during the time frame of the project to obtain candidates in a fair amount and/or in fair purity, compatible with further pharmacological explorations. Thus, only the lactam and the triazole bridges were explored and exemplified herein. Nevertheless, many difficulties linked to the solid phase synthesis strategies had to be overcome. Beside some more S-S bridge containing peptides, two series of pseudopeptides were obtained with potentially higher stability in biological milieus. These various pseudopeptides, analogues of the S38151 antagonist, were obtained in a sufficient amount, permitting them to be considered in a series of biological characterizations. Their sequences and main purity characteristics are shown in Appendix A.

### 2.2. Peptide Molecular Pharmacology

As in the past, our strategy was to measure the affinity of the peptides at the human MCH-R1 utilizing a classical binding assay [26]. The goal was to progress through various synthetic strategies until a low nanomolar compound was obtained. The antagonistic potential of these compounds was then evaluated. Due to the antagonism of S38151, it was expected that this antagonism remained in the newly synthesized pseudopeptides. Examples of the lactam and triazole bridges are given in Figure 1 to help the reader to visualize the different constructs.

#### 2.2.1. S-S Bridge Bearing Pseudopeptides

While the data were obtained simultaneously, we chose to present our progress by showing, first, the data for pseudopeptides bearing an S-S bond, as in the natural hormone as well as in our reference compound, S38151. The data are gathered in Table 1. Although we believed that the main problem would be the stability of the compound (see below), we made little progress in this series because only by changing Arg into hArg we found a slightly better candidate, based on its affinity at the MCH receptor.

#### 2.2.2. Lactam Bridge Bearing Pseudopeptides

At the same time, we explored the possibility of substituting the S-S bridge with a lactam one. This was performed by incorporating in position “2” an amino acid bearing a COOH or an NH2 at the end of their side chain, while the opposite was performed at position “11”. Note that the numbering is based on the natural sequence of the human MCH peptide (hMCH), even though amino acids were sometimes suppressed from the peptide sequences [26]. The data are shown in Table 2. As can be seen, a considerable number of analogues have been synthesized and characterized (71 different structures were studied). Incidentally, the linear peptides (e.g., GPS13670) were poor binders at the hMCH receptor. We tested a series of couples, including Asp or Glu for the acidic ones and Dap, Orn or Lys for the basic ones. We also, in some cases, tested the direction of the CO-NH moiety by including some NH-CO bridges (exchanging the acidic amino acids at position 2 by basic amino acids and vice versa). A handful of compounds (GPS13684, GPS13680, GPS13673, GPS12744, GPS13663, GPS13689, GPS15288, GPS13675, GPS13682 and GPS13683) presented sub-nanomolar affinities at the receptor.

Most efforts were put into this series of derivatives because we thought they were easier to obtain and the lactam bridge could lead to better stability. Two criteria directed our efforts: (i) the incorporation of exotic amino acids to enhance the diversity of the pseudopeptides, and (ii) the size of the ring of the cyclic peptides. Once again, the syntheses of several linear analogues demonstrated that the cyclic structure was important for the binding at the receptor. Nevertheless, it was a surprise to find that the linear compounds (GPS13670, GPS13669, GPS14523, GPS14489, and GPS14522) were not completely inactive on the target, although we did not pursue this path further (see, for instance, GPS 13670 and 13669 that bind at the hMCH with a Km in the 100 to 200 nM range).

Of note, only two syntheses of the peptides with the “inverse” configuration (acidic chain in position 11 and basic chain in position 2) were attempted: GPS12742 and GPS12745. Because they showed no major differences with their counterparts, further attempts were not performed, and the mainstream strategy remained with the acidic side chain at position 2 and the basic side chain at position 11. One way to modify or adjust the size of the ring (n’ being the number of atoms of the ring, see Figure 1) was to use various lengths of side chains in either category: Glu or Asp for acidic ones, and Orn, Lys, Dab or Dap for the basic ones. The goal was to keep the size within the 29 atoms length that seemed to be, with the sole exception of GPS13695 (where n’ is 32), the right size to fit the hMCH-R1 topology. Incidentally, attempts were made to crystalize these peptides to gain information on their structure, but these mostly failed. Only one piece of information seems to exist on the structure of MCH-derived peptide (S38151), which we previously described by NMR [28]

#### 2.2.3. Triazole Bearing Pseudopeptides

We then moved to a strategy in which we attempted to replace the S-S bridge by a triazole moiety, with the induced change in the general bulkiness of the bridge as compared to the initial disulfide bridge. The data are presented in Table 3. Once again, as we progressed during the various attempts during this exploration, we found very active compounds with affinities at the human receptor at least 10 times superior to S38151. For example, GPS15364 and GPS15366 have Km’s in the 200 pM range. Note also that the length of the crown that represents the cyclic pseudopeptide was maintained uniformly in this series at 29 atoms, as in S38151. Of note is also that the orientation of the triazole bridge (whether closer to the Gua terminus or the Trp terminus) seemed to favor an orientation where the triazole was on the Trp side, as can be seen for the three last peptides of Table 3. Linear peptide counterparts showed less potency than their cyclic analogues, as for GPS13683, bearing a canavanine at position 9, that presents an affinity of 1 nM at the receptor, while its linear counterpart, GPS14489 (see Table 2), is 1000 times less potent.

According to previous findings, all MCH analogues deprived of glycine at the position 5 found in the natural hormone were showing antagonistic activities. Indeed, starting at GPS12744, we found potent antagonistic compounds, as summarized in Table 4. Although antagonistic activity was measured on a single test, without considering the possible biasism [36] of the MCH system (as discussed for another receptor system [37,38]), we were confident from our previous knowledge of this receptor, that this antagonism would be translated in a living system like for S38151 [29].

As a conclusion of this section, after having synthesized about 150 peptides, several were found with (i) affinities at the receptor in the nM range and (ii) antagonism in the low pM ranges. Several of them were chosen to enter preliminary metabolic stability tests.

### 2.3. Stability of Some of the Pseudopeptides

Five pseudopeptides were selected to evaluate their stability in plasma from three species: mouse, rat and human. The various peptides were chosen because they represented, considering their internal bridges, the different families that we built, regardless of their performance in the binding assay. Table 5 shows the results. Globally, GPS15290 was the most stable compound. It was cyclized through a triazole bridge, placed between two amino acids incorporated in place of the two cysteines in the original antagonist, S38151, or in the natural hormone hMCH. Under these conditions, all the tested compounds were more stable than the initial antagonist, S38151.

The next step was to study in more detail the in vitro metabolism of the best compound in our hand, GPS15290. A metabolite identification study was then performed on rat and mice hepatocytes. GPS15290 demonstrated mild stability on hepatocytes of both species, although with an initial stage of disappearance probably attributable to the various possibilities of oxidation on the sulfur of the Met^3^. Metabolite comparative quantification showed some interspecies differences, while the global metabolic pattern observed (Figure 2) appeared to be quite complex.

As the response of the metabolites in tandem mass spectrometry (MS/MS) analytical systems is not known (synthetized metabolites were not available for calibration), the only quantitative comparison possible is between both species for each individual metabolite (expressed here as the relative percentage across species). At least 13 metabolites were measured under these conditions. They could be divided into two different classes: cyclic peptides with or without truncation and oxidation (M6, M8, M9, M17 and M19) and linear peptides because of the hydrolytic ring opening with or without truncation and oxidation (all others). The structure of one metabolite (M11) was not determined. All metabolites were observed in rats and mice except for M13, which was only observed in rats. The unchanged compound was measured at the end of the experiment.

### 2.4. From GPS15290 to GPS18169

As we expected problems with the GPS15290 compound due to the possible various oxidation states of the sulfur of Met^3^, we decided to also test a close analogue of this compound, GPS18169, in which Met was substituted by Nle. This compound has the same global features as GPS15290: particularly the number of atoms in the ring, and the length of a side chain of Nle compared to Met, but no sulfur in the side chain. As foreseen, the pseudopeptide maintained a nanomolar affinity at the receptor (Table 3) and an antagonistic functionality in the high picomolar range (see Table 4). Figure 3 presents four typical isotherms for both compounds (GPS15290 and GPS18169) in MCH-R1 binding and in cellular functional antagonism experiments.

GPS18169 showed similar stability in plasma, as compared to GPS15290 (Table 5). We measured about 50% of the initial dose remaining after incubation in mice plasma in the case of GPS18169, while GPS15290 remained in the low 20% range. With these data on GPS18169 at hand, the compound entered an in vivo experiment in the classical obesity model of high fat regimen.

### 2.5. In Vivo Activity of GPS18169 in the Diet-Induced Obesity Model in Mice

The model was classically established as detailed in the experimental section. In brief, four groups of mice were considered: a high-fat diet (HFD) (60% of calories) fed group and a standard chow diet (STD) fed group from 4 weeks of age for 8 weeks before i.p. vehicle treatment for 12 weeks, and two groups of high-fat diet (HFD) fed and administered either 5 or 10 mg/kg of GPS18169 by daily intraperitoneal injections for 12 weeks. The treatments did not lead to any significant effect on serum uric acid (UA), creatinine, potassium (K^+^) and sodium (Na^+^) levels when compared to the vehicle group on Week 16 (Day 113) and Week 20 (Day 141) (data not shown). On the contrary, variations were recorded considering the insulin, triglyceride levels and total cholesterol (see Table 6). A significant reduction in insulin was found after 12 weeks of treatment and, remarkably, the insulin level was back to the control, vehicle treated/normal diet when the mice were treated daily with 10 mg/kg of GPS18169 (grey cell, Table 6). Cholesterol and triglycerides levels remained high, despite a modest reduction during the treatments.

Compared to the vehicle control group, intraperitoneal injection of GPS18169 at 5 and 10 mg/kg once daily for 12 weeks was not associated with any significant effect on serum aspartate aminotransferase (AST) and alanine aminotransferase (ALT) levels on day 113 and day 141.

The weight of the five adipose tissues, including epididymal, mesenteric, retroperitoneal, inguinal, and brown fat, from C57BL/6 mice fed a high-fat diet was significantly (*p* < 0.05) increased compared to the mice fed a standard chow diet (STD). Intraperitoneal injection (IP) of GPS18169 at 5 and 10 mg/kg once daily for 12 weeks showed a significant (*p* < 0.05) decrease in the weight of mesenteric, retroperitoneal, inguinal, and brown fat on Week 20 (Day 141). The weight ratio of adipose tissue to the whole body (g/100 g of body weight [BW]) was constant in the study (Table 7).

The food and water intake in the group of C57BL/6 mice fed a high-fat diet was significantly (*p* < 0.05) decreased during the study period when compared to the group of mice fed a standard chow diet (STD). However, the food and water intake between the vehicle and test article treated groups was similar during the study period (Appendix A and Figure 4 A,B).

Moreover, body weight gain in the group of C57BL/6 mice fed a high-fat diet during the study period between Week 4 (Day 29) and Week 20 (Day 141) was significantly (*p* < 0.05) higher than that in the group of mice fed a standard chow diet (STD) (Figure 5). Intraperitoneal injection (IP) of GPS18169 at 5 and 10 mg/kg once daily for 12 weeks showed a significant (*p <* 0.05) decrease in body weight gain from Day 117 to Day 141 when compared to the vehicle group that were similarly injected once daily with the vehicle (Figure 5). Neither overt toxicities nor behavioral modifications were observed in the treated animals in the study, while some behavioral concerns have been reported with other MCH-R1 antagonist treatments, such as for TASP0382650 [39], ATC0065 [40] GW803430 [41] or SNAP-7941 [42] (among others, see [43,44]).

## 3. Discussion

Obesity, and subsequent complications including, but not limited to, diabetes, is a world-scale problem, enhancing the death rate and the cost of maintaining the population in good health. Because of this, since the late 1980s, it was declared a priority to find ways to fight obesity. In general, two types of obesity could be distinguished: (i) the genetic one [45], such as leptin deficiency [46], in which mutations can lead to the over-accumulation of fat or, in some cases, linked to pharmacological treatments such as anti-HIV therapies [47], and (ii) the obesity linked to the lack of control of appetite, as well as universal access to food regimens over-rich in both fat and sugar (commonly called cafeteria regimens [48] or Western diet [49]).

Although the goal of the present work was not to discuss the structural aspects of the molecular pause of the antagonist(s) onto the receptor, some observations are worth mentioning, without going into more technological approaches such as molecular modeling. Originally, the natural agonist, MCH, was engineered to bring in several unnatural features to introduce higher stability and patentability, and to obtain antagonist(s). At first, we identified a minimal binding sequence from MCH. It resulted in the peptide from which five N-terminal amino-acids [Asp-Phe-Asp-Met-Leu] and two of the C-terminal side [Gln-Val] were deleted [26]. The final product was a shorter agonist with an affinity at the MCH-R1 receptor superior to the natural peptide [26]. To reach that goal, we produced more than 200 analogues with, early on, the change of the N-terminal amino acid, arginine, by a p-guanidino benzoic acid. In doing so, the N-terminal amino group was removed, and with it, the possibility for the peptide to be catabolized by exo-aminopeptidase. On the contrary, the C-terminal amino-acid, tryptophan, could not be substituted so radically, and only close analogues, such as 3-benzothienylalanine, were introduced. We turned the agonist into an antagonist by deleting the Gly in position 10 of the natural hormone [28], leading to S38151. This compound served as a starting point for the present investigation with the goal to replace the S-S bridge, synonymous of poor stability in vivo, with a more stable version of the antagonist. Interestingly, our initial attempts to produce the cyclic part of the peptide S38151 (including a cystine bridge) led to completely inactive molecules [26]. We realized rapidly that the tyrosine (at position 13 of the original hormone) was not necessary to sustain the activity of the corresponding peptide, a feature that escaped our attention in our previous explorations [26,28]. Our main surprise was the observation that, despite the slight hindrance of the triazole moiety (when compared to the disulfide bridge) introduced in the bridge of the pseudopeptide, the best peptide belongs to this click-generated family of compounds. Because we tried to substitute the S-S bridge, we wrongly reasoned that a rather flat moiety deprived of chemical cycles would create a better balance between stability and affinity/activity. GPS18169, on the contrary, bears a penta-atom cycle in the bridge. Furthermore, the combination of the size of the cycle and the different amino acids at different positions in the sequence was obvious. For example, inside the lactam family of analogues sharing the same cycle size resulting from the incorporation of the same acid/base side chains (Table 4), several “mutations” led to compounds with Ki below 1 nM, although the differences at this level of activity/affinity should be taken cautiously. Similarly, certain compounds coming from the triazole family showed that the affinity of the initial compound could be “ameliorated”, for example, GPS15363. To maintain a 29-atom ring as in the initial antagonist S38151, we had to compensate for the triazole bridge that would count for seven atoms. The suppression of Tyr^8^ in the sequence made for a three heavy atom compensation, and the last ones were compensated as follows: at the position “7”, a beta-amino acid, beta-homovaline, was incorporated instead of the original Val. It is interesting to note that the size of the triazole should account for more than three atoms, as a supplementary compensation was necessary to recover the potent activity of the initial compound.

Initially, we started by testing GPS18169 in a genetic model of obesity (fa/fa rats) related to a defect in the leptin receptor. The experiments failed as no changes were recorded between treated and untreated animals (data not shown), unsurprisingly suggesting that leptin-independent pathway(s) exist(s) in the pathophysiology of genetic obesity that does not involve the MCH pathway. A leptin-independent animal model should be used to confirm the implication of the MCH pathway and the effect of receptor antagonist(s) in regulating food intake and weight control. To document the properties of these antagonistic putative peptides derived from MCH, we used the established diet-induced obesity model [50,51], as it reflects the situation of the overweight/obesity pandemic represented in most Western countries [52], and beyond [53]. Peptides such as D-Ala2-Met-enkephalinamide [54] or NPY5 [55] and MCH were among the targets explored in the past to attempt to control appetite. The MCH-R1 antagonists might very well fill the task, as they seem to act not only at the central level [42] but also peripherally [56], maybe by enhancing the energetical use of fat.

On the physio-pathological side, although without digging into the details of the possible mechanism(s) of action of the antagonist downstream its receptor, one should point out several points. First, due to the affinity of the GPS18169 at its receptor, in the low nanomolar range, and its pseudopeptidic nature, one could be convinced that the compound is rather specific to its target, a feature always difficult to prove and obtain with small molecules. Second, the cyclisation by click chemistry could be an avenue to stabilize naturally occurring cyclic peptides with an original S-S bridge, and cyclic peptide examples are numerous (see, for example, Bockus et al. [57]). Incidentally, these authors stressed that “naturally occurring cyclic peptides exhibit a wide variety of unusual and potent biological activities” and “penetrate cells by passive diffusion”, which significantly point to the necessity to exploit more from natural sources, even peptides, the structure of which might seem complex at first glance (see also discussion in Boutin et al. [35]). Third, the decrease observed in adipose tissue mass in the treated animals also concurs with the widely admitted fact that adipose tissue has a role as an endocrine regulator of insulin resistance [58,59]. It is concluded that GPS18169, given at 5 and 10 mg/kg intraperitoneally once daily for 12 weeks, had significant protective effects in the study, and this effect was accompanied by an improvement in body weight gain and serum insulin, total cholesterol (TC) and triglyceride (TG) levels. The GPS18169 treatment also showed a significant decrease in adipose tissue weight in a high-fat diet-induced obesity model in C57BL/6 mice.

This program started more than 20 years ago [26]. The final stages were designed to answer three questions: Would it be possible to extrapolate from our acquired knowledge on MCH antagonists [26,28] peptide or pseudopeptidic structures that would be stabilized, as compared to our previous candidate, S38151? Would they be as potent as the previous peptides in our molecular pharmacology assay? Would the best candidate be active in one in vivo assay? Using this new compound, we clearly answered the three questions: natural cyclic peptides could be stabilized using alternative bridge approaches, without changing their affinity at the receptor. These observations and consequent syntheses of pseudopeptides led to a compound with interesting anti-obesity (or anti-obesity-leading complications, such as insulin decrease) features. If the latter observation is limited to the MCH system and obesity, the two former answers have a more general exemplification role, such as how to manipulate peptide chemistry to obtain accurate tools (or drugs?).

## 4. Materials and Methods

### 4.1. General Information

Standard Fmoc-amino acids, Oxyma Pure, *N,N′*-diisopropylcarbodiimide (DIC), Fmoc-L-hArg(Pbf)-OH (CAS N°: 1159680-21-3), Fmoc-D-Arg(Pbf)-OH (CAS N°: 187618-60-6), Fmoc-5-Ava-OH (CAS N°: 123622-48-0), Fmoc-L-2-Abu-OH (CAS N°: 135112-27-5), Fmoc-L-Asp(OAll)-OH (CAS N°: 146982-24-3), Fmoc-L-Dap(Alloc)-OH (CAS N°: 188970-92-5), Fmoc-L-Glu(OAll)-OH (CAS N°: 133464-46-7), Fmoc-L-Orn(Alloc)-OH (CAS N°: 147290-11-7), Fmoc-L-N-Me-Cys(Trt)-OH (CAS N°: 944797-51-7), Fmoc-L-beta-hMet-OH (CAS N°: 266359-48-2), Fmoc-L-ethionine (CAS N°: 1562431-51-9), Fmoc-L-buthionine (CAS N°: 1821797-31-2), Fmoc-L-Selenomethionine (CAS N°: 1217852-49-7), Fmoc-L-Pra-OH (CAS N°: 198561-07-8), Fmoc-L-Aha-OH (CAS N°: 942518-20-9), Fmoc-L-Nle-OH (CAS N°: 77284-32-3), Fmoc-L-Tle-OH (CAS N°: 132684-60-7), Fmoc- Aib-OH (CAS N°: 94744-50-0), Fmoc-L-Arg(NO2)-OH (CAS N°: 58111-94-7), Fmoc-L-Arg(Me,Pbf)-OH (CAS N°: 1135616-49-7), Fmoc-L-Cit-OH (CAS N°: 133174-15-9), Fmoc-L-Cav(Boc)-OH (CAS N°: 190723-97-8), Fmoc-L-Pro(4-NHBoc)-OH (CAS N°: 221352-74-5), Fmoc-L-5,5-di-Me-Pro-OH (CAS N°: 1310680-23-9), Fmoc-L-Pro(4-CF3)-OH (CAS N°: 1242934-32-2), Fmoc-L-Pro(4-Ph)-OH (CAS N°: 269078-71-9), Fmoc-4-Abz-OH (CAS N°: 185116-43-2), Fmoc-L-Lys(N3)-OH (CAS N°: 159610-89-6), Fmoc-L-beta-hVal-OH (CAS N°: 172695-33-9) and WANG resin (200–400 mesh, 0.34 mmol/g) were purchased from Iris Biotech. Fmoc-5,5,5-trifluoro-DL-Leu-OH (CAS N°: 777946-04-0), Fmoc-4,5-dehydro-L-Leu-OH (CAS N°: 87720-55-6), Fmoc-4-hydroxy-L-Leu-OH, and Fmoc-Bta-OH (CAS N°: 177966-60-8) were purchased from Eurogentec. Fmoc-L-Hpra-OH (CAS N°: 942518-21-0), Fmoc-N-Me-L-Met-OH (CAS N°: 84000-12-4), and Fmoc-N-Me-L-Trp-OH (CAS N°: 112913-63-0) were purchased from ChemPep. Dimethylformamide (DMF), CH_2_CL_2_, methanol, diethyl ether, and HPLC-grade acetonitrile were purchased from Sigma-Aldrich, St Louis, Mo, USA.

### 4.2. Peptide Synthesis

All peptides were synthesized at 100 μmole scale using Fmoc-SPPS on a SymphonyX instrument (Gyros Protein Technologies, Uppsalan Sweden). The standard deprotection-coupling cycle for each residue consisted of six steps: 1. Wash with 5 mL of DMF (3 × 30 sec); 2. Fmoc group deprotection with 5 mL of 20% piperidine in DMF (3 × 3 min); 3. Wash with 5 mL of DMF (3 × 30 sec); 4. Fmoc-AA-OH coupling (2 × 60 min); 4a. 5 mL of 0.2M Fmoc-AA-OH in DMF; 4b. 2 mL of 0.5M OxymaPure in DMF; 4c. 1 mL of 1M DIC in DMF; 5. Capping with 5 mL of 10% Ac2O in DMF (7 min); and finally 6. Wash with 5 mL of DMF (3 × 30 sec).

Upon completion of the synthesis, peptide resins were washed 3× with DMF and 3× with CH_2_CL_2_. Peptides were then cleaved with one of three trifulroacetic acid (TFA) cleavage cocktails (per 100 μmole scale synthesis): (a) Cocktail #1: Peptides containing neither Met nor Cys: 18 mL TFA, 500 μL triisopropylsilane (TIS), 500 mg phenol, 1000 μL water; (b) Cocktail #2: Peptides containing Cys, but no Met: 17.5 mL TFA, 500 μL TIS, 500 mg phenol, 1000 μL water, 500 μL EDT; or (c) Cocktail #3: Peptides containing Met: 17.5 mL TFA, 500 μL TIS, 500 mg phenol, 1000 μL water, 500 μL EDT and 500 µL tetrabutylammonium bromide. After 3h of TFA cleavage, the cleavage solution was filtered from the resin, and peptides were precipitated by addition to 40 mL of ice-cold diethyl ether. After >1 h at −20 °C, ether solutions were centrifuged at 3500 rpm, and the supernatants were decanted. Pellets were washed twice with diethyl ether and then dried for >3 h in a vacuum desiccator before dissolution, analytical characterization, and purification. The peptides were analyzed by Ultra Performance Liquid Chromatography (UPLC) and Electrospray Ionization Mass Spectrometry (ESI-MS). The instruments were equipped with BEH C18 (Waters), 150 mm × 2.1mm (flow rate: 0.6 mL/min). Solvents A and B were 0.1% TFA in water and 0.1% TFA in acetonitrile. One hundred and three peptides were synthesized in this project.

#### 4.2.1. Synthesis of Peptides with Lactam Bridge

Peptides were prepared by standard Fmoc-SPPS as described above, using protected amino acids. Upon completion of the linear peptide synthesis, the Alloc group was removed. PhSiH3 (25 eq.) in 5 mL of CH_2_Cl_2_ was added to the resin followed by Pd(PPh3)4 (0.25 eq.) in CH_2_Cl_2_. After 1h resin agitating in the dark, the solution was drained and the reaction was repeated. The resin was washed with 3 volumes of CH_2_Cl_2_, 4 volumes of 1M pyridine in DMF and 3 volumes of DMF. OxymaPure (3 eq.) and DIC (3 eq.) in 5 mL of DMF were added to the resin and agitated overnight. The resin was washed with 3 volumes of DMF and 3 volumes of CH_2_Cl_2_. After standard peptide cleavage conditions (described above), the crude peptide was purified by Preparative HPLC and lyophilized.

#### 4.2.2. Synthesis of Peptides with a Disulfide Bridge

Peptides were prepared by standard Fmoc-SPPS as described above, using protected amino acids. After standard peptide cleavage conditions (described above), the crude peptide was dissolved in 200 mL of acetic acid solution (AcOH/H_2_O 1:9), then a solution of 0.06 M iodine in MeOH was added dropwise with rapid stirring until the solution became slightly yellow. The excess of iodine was quenched with 1 M ascorbic acid and the crude peptide was purified by Preparative HPLC and lyophilized.

#### 4.2.3. Synthesis of Peptides with Triazole Bridge on a Solid Support

Peptides were prepared by standard Fmoc-SPPS as described above, using protected amino acids. Upon completion of the linear peptide synthesis on a solid support, CuI (2 eq.), sodium ascorbate (2 eq.) and *N, N*-diisopropylethylamine (DIPEA) (3 eq.) in 5 mL of *N*-methyl pyrrolidone were added to the resin. After 1h agitation of the resin under an inert atmosphere, the solution was drained and the resin was washed consecutively with 0.5% DIPEA in DMF, 0.5% (*w/w*) sodium diethyldithiocarbamate in DMF and 1M pyridine in DMF and DMF to remove the catalyst. After standard peptide cleavage conditions (described above), the crude peptide was purified by preparative HPLC and lyophilized.

### 4.3. Purification and Analyses

Purifications of crude peptides were performed with a preparative reversed phase HPLC (Waters Delta Prep 4000, Waters Corp, Milford, MA, USA) system using a reversed phase column (Vydac Denali prep C-18, 10 µm, 120 Å, 50 × 300 mm) (VWR, Radnor, PA, USA) and an appropriate gradient of increasing concentration of buffer B in buffer A (flow rate of 80 mL/min). The fractions containing the purified target peptide were identified by UV measurement (Waters 2489 UV/Visible detector) at 214 nm and selected fractions were then combined and lyophilized. 

A summary of the peptide sequences is presented in Appendix A, while their analytical characteristics are shown in Appendix A. Peptides were numbered according to Genepep internal system (GenePep Synthesis).

### 4.4. Establishment of a Stable Cell Line Expressing Human MCH Receptor

Stable cell lines were obligatory for all receptor programs. We cloned the human receptor and established the cell line as previously described [26,27]. In brief, HEK293 (HEK) cells were grown in Dulbecco’s Modified Eagle medium (DMEM) supplemented with 10% fetal calf serum, 2 mM glutamine, 100 IU/mL penicillin and 100 mg/mL streptomycin. They were seeded at 5 × 10^6^ cells in T75 cm^2^ culture flasks. One day later, they were transfected with 10 mg of the pcDNA3.1 (Invitrogen, Groningen, The Netherlands) containing the human MCH-R1 using lipofectamine (Life Technologies, Cergy Pontoise, France), as previously described. One day later, cells were trypsinized, resuspended in a complete DMEM medium containing 800 mg/mL of geneticin and seeded at different dilutions in 96-well plates which were kept for 2 to 3 weeks in a humidified CO_2_ incubator. At the end of the selection period, isolated clones were picked up, amplified, and further characterized by cyclic AMP experiments. For each cell line, one positive clone was subcloned before being used for all of the cyclic AMP, [^35^S]-GTPγS and receptor binding experiments necessary to characterize them [26]. The stable HEK293 cell line expressing the human MCH-R1 was HEK-hMCH-R1.

### 4.5. Membrane Preparation and Binding

Membrane preparations were classically made as previously described [26,27]. In brief, the HEK-hMCH-R cell line was grown to confluency, harvested in PBS buffer containing 2 mM EDTA and centrifuged at 10,000× *g* for 5 min (4°C). The resulting pellet was suspended in 20 mM Hepes buffer (pH 7.5), containing 5 mM EGTA and homogenized using a Kinematica polytron. The homogenate was then centrifuged (95,000× *g*, 30 min, 4 °C) and the resulting pellet was suspended in 50 mM HEPES buffer (pH 7.5), containing 10 mM MgCl_2_ and 2 mM EGTA. Aliquots of membrane preparations were stored at −80 °C until use. Binding has been described in numerous publications and the test was based on Mac Donald et al.’s description [60]. Cell membrane homogenates (4 μg protein) are incubated for 60 min at 22 °C with 0.1 nM [^125^I]-[Phe^13^,Tyr^19^]-MCH in the absence or presence of the test compound in a buffer containing 25 mM Hepes/Tris (pH 7.4), 5 mM MgCl_2_, 1 mM CaCl_2_ and 0.5% bovine serum albumin. Nonspecific binding is determined in the presence of 0.1 μM MCH. Following incubation, the samples are filtered rapidly under vacuum through glass fiber filters (GF/B, Packard) presoaked with 0.5% BSA and rinsed several times with an ice-cold buffer containing 50 mM Tris-HCl (pH 7.4) and 500 mM NaCl using a 96-sample cell harvester (Unifilter, Packard). The filters are dried and then counted for radioactivity in a scintillation counter (Topcount, Packard) using a scintillation cocktail (Microscint 0, Packard). The results are expressed as percentage inhibition of the control radioligand specific binding. The standard reference compound is MCH, which was tested in each experiment at 8 concentrations to obtain a competition curve from which its IC_50_ is calculated. The high level of assay robustness and reproducibility of each control monitored over several decades (internal data base, Eurofins) and included in every assay ensure results’ reliability even when performed only twice. Experimental points were obtained in two independent measurements, each in duplicate. All data were in less than 5% variation from each other or they were discarded and re-ran until this variation was reached. Whenever IC50 was concerned, a difference between the two results of more than half a log invalidates the experiment that will be re-run.

### 4.6. MCH Functional Cellular Assay: MCH-Induced Cytosolic Ca^2+^ Ion Mobilization

The functional assay was run onto cells expressing the hMCH-R1 receptor, as described in Wang et al. [61]. In brief, the cells are suspended in DMEM buffer (Invitrogen), then distributed in microplates at a density of 4.104 cells/well. The fluorescent probe (Fluo4 Direct, Invitrogen) mixed with probenecid in Hanks’ Balanced Salt solution (HBSS) (Invitrogen) complemented with 20 mM HEPES (Invitrogen) (pH 7.4) is then added into each well and equilibrated with the cells for 60 min at 37 °C, then 15 min at 22 °C. Then, the assay plates are positioned in a microplate reader (CellLux, PerkinElmer) which is used for the addition of the test compound, reference agonist or HBSS buffer (basal control), and the measurements of changes in fluorescence intensity, which varies proportionally to the free cytosolic Ca^2+^ ion concentration. For stimulated control measurements, MCH at 300 nM is added in separate assay wells. The results are expressed as a percentage of the control response to 300 nM MCH. The standard reference agonist is MCH, which is tested in each experiment at several concentrations to generate a concentration-response curve from which its EC_50_ value is calculated. For the antagonist effect, the results are expressed as a percentage inhibition of the control response to 30 nM MCH, which was added to all tested wells and controls 5 min after the candidate products.

### 4.7. Stability of Pseudopeptides, Plasma and Microsomes from Mouse, Rat and Man

#### 4.7.1. Plasma Stability

Plasmas from 3 species (including commercially available human healthy donors) were adjusted to pH 7.4 if necessary. Incubations were performed at a concentration of 1 µM of the compound at 37 °C (in the presence of a final concentration of DMSO of 2.5% *v/v*. The incubations were stopped at 0, 15, 30, 60 and 120 min by methanol. Each time point was run in sextuplets. The sampling plates were then centrifuged (2500 rpm, 45 min, 4 °C) and the supernatants pooled. Samples were analyzed for parent compound by LC-MS/MS. The percentage of parent compound remaining at each time point relative to the 0 min sample is then calculated from LC-MS/MS peak area ratios.

#### 4.7.2. Mouse and Rat Hepatocytes

The stability of the compound was studied on hepatocyte cultures from Wistar rats and CD1 mice. The compound was incubated at 20 µM in the presence of 1.106 cells/mL for 5 h. The incubation was then stopped with acetonitrile (*v/v*) and the supernatant, after a brief centrifugation (2500 rpm, 5 min 4 °C) was submitted to LC-MS analysis, using a THERMO Q Executive coupled to a Waters Acquity UPLC System equipped with a Waters Acquity UPLC C18 100 × 2.1 mm 1.7 µm column.

#### 4.7.3. Mouse, Rat and Human Hepatic Microsomes

Metabolic stability was determined at 100 nM on hepatic microsomes from mouse, rat and man (0.3 mg proteins/mL) from commercial sources. Experiments were performed kinetical, by sampling the medium at the following time points: 0, 5, 15, 30, 45 min. The incubation was stopped by mixing the sample (*v/v*) with pure acetonitrile. Control experiments were run in which NADPH was omitted. All incubations were performed singularly for each test compound but can be reproduced in case of criteria not being fulfilled. LC-MS/MS analysis of these samples will be performed on the most sensitive instrument.

### 4.8. Evaluation of Metabolic Stability of the Candidate Pseudopeptides: GPS15290 and GPS18169

Historically, the first compound in our collection to show interesting characteristics was GPS15290, with a superior affinity at the human receptor and an antagonistic capacity in the high picomolar range. We thus studied its metabolic stability in rats and mice. 

Then, the stability in plasma was evaluated by incubating human, rat or mouse plasma with the compound(s) as follows: the product (0.1 µM in TRIS buffer pH 7.4 0.1% BSA) was incubated with the plasma at 37 °C. The disappearance of the product was measured at 15, 45, 120, 240 and 300 min by mass spectrometry. Incubations were carried out in a robot Star (Hamilton France, Villebon-sur-Yvette, France). At each sampling time, 50 µL of the reaction medium was removed and diluted with 100 µL of acetonitrile. After centrifugation (20 min at + 4 °C, 4000 rpm), 75 µL of the supernatants was transferred into a 384 well injection plate. Ten microliters were injected onto UPLC coupled with a XEVO TQS mass spectrometer system (Waters, Milford, MA, USA). The unchanged product is quantified using a reference sample (0.1 µM) prepared during the incubation process. A second reference sample at 0.01 µM is prepared in the same conditions to verify the linearity of the detection method. The peak height (or area) is measured to calculate the half-life of the compound.

### 4.9. Diet-Induced Obesity Experiments

These experiments were performed by Pharmacology Discovery Services Taiwan, Ltd. (New Taipei City, Taiwan), a subsidiary of Eurofins Scientific (Luxembourg, Luxembourg) for the Institut de Recherches Internationales Servier.

#### 4.9.1. Animals

Male C57BL/6 mice at 4 weeks of age were provided by BioLasco Taiwan (under Charles River Laboratories Licensee). All animals were maintained in an environment with well-controlled temperature (20–24 °C) and humidity (30–70%) with 12 h light/dark cycles. Free access to standard lab diet (MFG (Oriental Yeast Co., Ltd., Japan)) and autoclaved tap water was granted. All aspects of this work including housing, experimentation, and animal disposal were performed in general accordance with the “Guide for the Care and Use of Laboratory Animals: Eighth Edition” (National Academies Press, Washington, D.C., 2011) in our laboratory animal facility accredited by the International Council on Accreditation adopted the AVMA Guidelines for the Euthanasia of Animals (AAALAC). In addition, the protocol for animal care and use was reviewed and approved by the Institutional Animal Care and Use Committee (IACUC) at Pharmacology Discovery Services Taiwan, Ltd.

#### 4.9.2. Origin of Biochemical Tests 

ALT (Alanine aminotransferase) and AST (Aspartate aminotransferase) assay kit (Denka Seiken, Japan), Cholesterol assay kit (Denka, Japan), Creatinine assay kit (Denka Seiken, Japan), Mouse insulin ELISA kit (Crystal chem, Elk Grove Village, IL, USA), Na^+^/K^+^ assay kit (Toshiba, Japan), Triglyceride assay kit (Sentinel Diagnostics, Milano, Italy) and Water for injection (Tai-Yu, Taiwan).

#### 4.9.3. Diet-induced Obesity Protocol

The protocol was performed for all groups and all animals. Male C57BL/6 mice were fed a high-fat diet (HFD) (60% of calories) or a standard chow diet (STD) from 4 weeks of age. Thirty (30) mice were fed the HFD and 10 mice were fed the STD (for 8 weeks before treatment) for a total of 20 weeks. The mice assigned to the STD diet were maintained on this diet throughout the study as a reference group of lean control mice. From Week 9, HFD-fed mice were grouped by body weight before dosing. Body weight was recorded three times weekly from Week 9. Food/water intake was recorded twice weekly from Week 9. Blood chemistry analyzes were performed in Weeks 8 (before dosing, Day 56), 16 (Day 113) and 20 (Day 141), including serum insulin, total cholesterol (T-CHO), triglyceride (TG), alanine aminotransferase (ALT), aspartate aminotransferase (AST), uric acid (UA), creatinine, potassium (K^+^) and sodium (Na^+^). At the termination in Week 20 (Day 141), the adipose tissue weight, including epididymal, mesenteric, inguinal, retroperitoneal, and brown fat, was measured after the animals were sacrificed. Two-way ANOVA followed by Bonferroni’s tests were applied to ascertain the difference between vehicle control and treated animals. Significance was set at *p* < 0.05 level. The summary of the treatments is presented in Table 8. The following parameters were measured: (1) body weight three times weekly; (2) food/water intake twice weekly; (3) blood chemistry in Weeks 8 (before dosing), 16 and 20 including glucose, insulin, serum lipids (triglycerides (TG), total cholesterol (TC), AST, alanine aminotransferase (ALT), uric acid, creatinine, potassium and sodium through 16-h fasting animals; and finally, (4) total body fat (including visceral and subcutaneous fat) in Week 20.

## Figures and Tables

**Figure 1 molecules-26-01291-f001:**
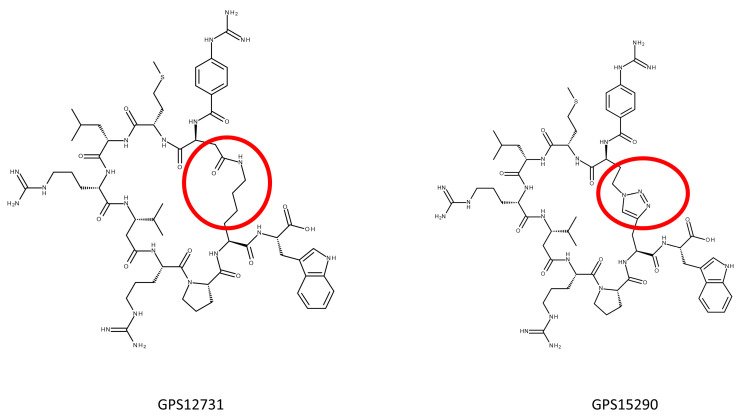
Two structures of peptides representative of the families bearing either a lactam bridge (GPS12731) or a triazole bridge (GPS15290). The lactam bridge was built between a glutamic acid (or aspartic acid for other pseudopeptides) and ornithine (or a lysine) side chains. The triazole bridge was synthesized between an azido-homoalanine and a homopropargylglycine. The red circles point at the bridge structure.

**Figure 2 molecules-26-01291-f002:**
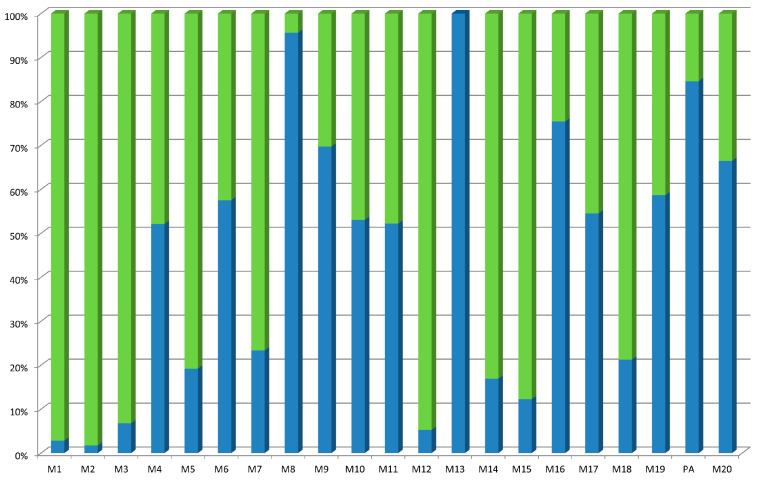
Comparative view of the relative intensity of GPS15290 metabolites in vitro in hepatocytes. Green: mouse metabolites, blue: rat metabolites. The starting pseudopeptide is GPS15290: pGua-[N_3_-hAla-MLR-βhVal-RP-hPra]-W-OH (mass = 1352.7); M1 is (hPra -W)-(N_3_-hAla-pGua) (mass = 604.2); M2 is R-βhVal-RP (mass = 540.3); M3 is (hPra-W)-(N_3_-hAla-pGua)-M (mass = 735.3); M4 is βhVal-RP-(hPra-W)-(N_3_-hAla-pGua) (mass = 970.5); M5 is P-hPra-(N_3_-hAla -pGua)-MLR-βhVal-R (mass = 1184.6); M6 is pGua-[(N_3_-hAla + O^-^)-(M+O)LR-βhVal-RP-hPra]-W-OH, oxidations on Met and on N_3_-hAla (mass = 1384.7); M7 is R-βhVal-RP-(hPra-W)-((N_3_-hAla + O^-^)-pGua), oxidation on N_3_-hAla (mass = 1142.6); M8 is pGua-[N_3_-hAla-MLR-βhVal-RP-hPra] (mass is 1166.6); M9 is pGua-[N_3_-hAla-(M+O)LR-βhVal-RP-hPra]-W-OH oxidation on Met (mass = 1368.7); M10 is R-βhVal-RP-(hPra-W)-(N_3_-hAla-pGua) (mass = 1126.6); M11, with a mass of 1179.5 remained unknown; M12 is (hPra-W)-(N_3_-hAla-pGua)-ML (mass = 848.4); M13 is LR-βhVal-RP-hPra-(N_3_-hAla-pGua) (mass = M1184.6); M14 is pGua-[N_3_-hAla-(M+O)LR-βhVal-RP-hPra]-W-OH, oxidation on Met and opening by hydrolysis (mass is 1386.7); M15 is pGua-[(N_3_-hAla + O^-^)-MLR-βhVal-RP-hPra]-W-OH oxidation on Met and opening by hydrolysis (mass = 1386.7); M16 is MLR-βhVal-RP-(hPra-W)-(N_3_-hAla-pGua) (mass is 1370.7); M17 is pGua-[(N_3_-hAla + O^-^)-MLR-βhVal-RP-hPra]-(W + O)-OH, oxidations on Met and on Trp (mass is 1384.7); M18 is pGua-[N_3_-hAla-MLR-βhVal-RP-hPra]-W-OH, opening by hydrolysis (mass is 1370.7); M19 is pGua-[(N_3_-hAla + O^-^)-MLR-βhVal-RP-hPra]-W-OH, oxidation on N_3_-hAla (mass = 1368.7); M20 is pGua-[N_3_-hAla-MLR-βhVal-RP-hPra]-W-OH, opening by hydrolysis (mass is 1370.7). These attributions were carried out based on ms/ms data.

**Figure 3 molecules-26-01291-f003:**
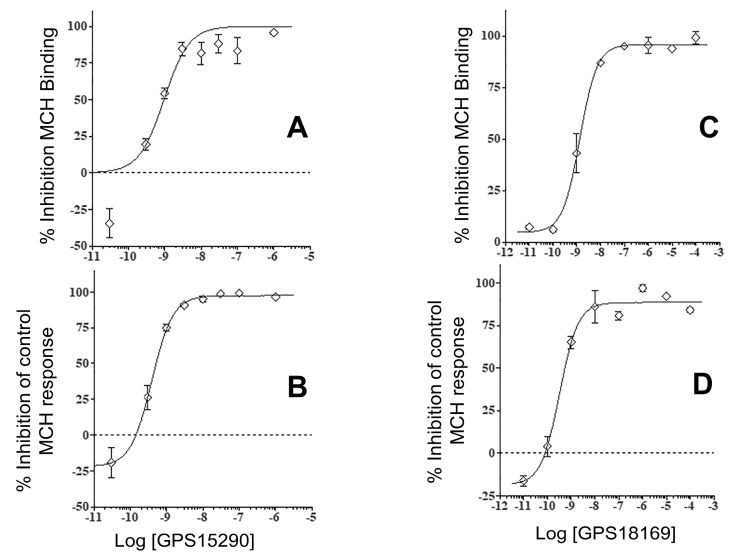
Examples of isotherms obtained with the two antagonistic pseudopeptides, GPS13290 and GPS18169. The assays were binding displacement experiments (panels **A, B**) of labelled MCH at membranes overexpressing the human MCH-R1 or MCH-induced cytosolic Ca^2+^ ion mobilization inhibition in living cells (panels **C, D**). The ligand was [^125^I]-[Phe^13^,Tyr^19^]-MCH. Independent experiments were performed at least twice using different batches of membranes from stably transfected CHO cells, and each point was obtained in triplicate. Concentration isotherms were obtained using eight concentrations of each product from 10–11 to 10–6 M. The data represent the mean ± SD of the triplicate measure. Experiments were run at least twice, independently.

**Figure 4 molecules-26-01291-f004:**
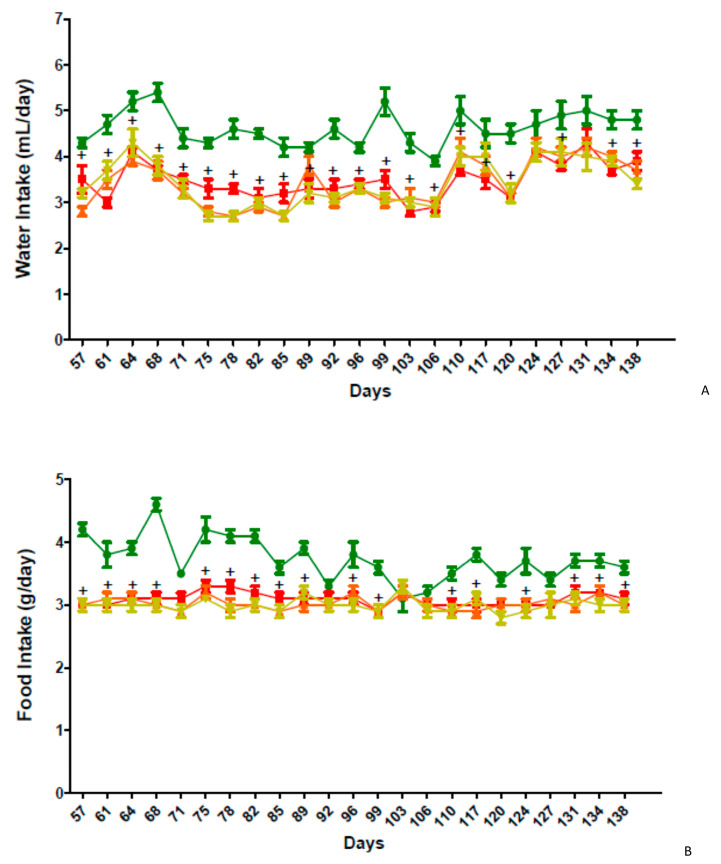
Water (**A**) and food (**B**) intake in high-fat diet-induced obesity model in C57BL/6 mice. Male C57BL/6 mice were fed a high-fat diet (HFD, 60% of calories) or a standard chow diet (STD) from 4 weeks of age for 8 weeks before treatment. Once obesity was installed, GPS18169, at 5 and 10 mg/kg and vehicle (WFI) was administered by intraperitoneal injection (IP) once daily for 12 weeks after 8 weeks of HFD feeding. Food (**B**) and water (**A**) intake were recorded twice weekly. ^+^
*p* < 0.05, vs. Vehicle (High-fat Diet); two-way ANOVA followed by Bonferroni test. Green squares: vehicle and normal diet; red squares: vehicle and high fat diet; orange triangles: GPS18169 (10 mg/kg) and high fat diet; greenish-yellow triangles: GPS18169 (5mg/kg) and high fat diet. Complete individual data are presented in Appendix A (Food).

**Figure 5 molecules-26-01291-f005:**
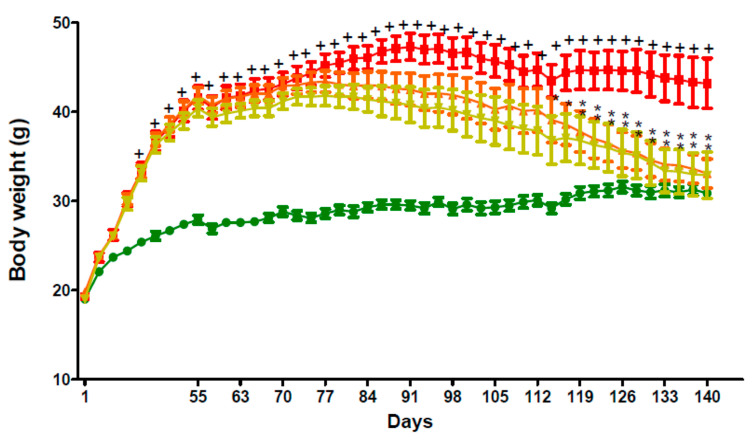
Body weight changes in high-fat diet-induced obesity model in C57Bl/6 mice. Male C57BL/6 mice were fed a high-fat diet (HFD) (60% of calories) or a standard chow diet (STD) from 4 weeks of age for 8 weeks before treatment. Test article, GPS18169, at 5 and 10 mg/kg and vehicle (WFI) were administered by intraperitoneal injection (IP) once daily for 12 weeks after 8 weeks of HFD feeding. The body weight was measured thrice a week. Complete individual data are presented in Appendix A (body weight changes).

**Table 1 molecules-26-01291-t001:** Binding affinities at human melanin-concentrating hormone (MCH) receptor of pseudopeptides derived from S38151, bearing an S-S bridge.

Code	Sequence												Km (nM) *	n’
	**1**	**2**	**3**	**4**	**5**	**6**	**7**	**8**	**9**	**10**	11	12		
S38151 (**)	Gua	Cys	Met	Leu	-	Arg	Val	Tyr	Arg	Pro	Cys	Trp	4,3	29
GPS11371	Gua	Cys	Met	Leu	-	Arg	-	Tyr	Arg	Pro	Cys	Trp	~100	26
GPS11372	Arg	Cys	Met	Leu	-	Arg	-	Tyr	Arg	Pro	Cys	Trp	>10,000	26
GPS11373	Gua	Cys	Met	Leu	-	Arg	Val	-	Arg	Pro	Cys	Trp	100	26
GPS11374	Gua	Cys	Met	Leu	-	Arg	Val	Tyr	hArg	Pro	Cys	Trp	3	29
GPS11375	Gua	Cys	Met	Leu	-	Arg	Val	Tyr	arg	Pro	Cys	Trp	17	29
GPS11376	Gua	Cys	Met	Leu	-	Arg	Val	Ala	hArg	Pro	Cys	Trp	~100	29
GPS11377	Gua	Cys	Met	Leu	-	-	Val	Tyr	Arg	Pro	Cys	Trp	27	26
GPS11378	Gua	Cys	Met	Leu	-	Ala	Val	Ala	hArg	Pro	Cys	Trp	18	29
GPS11379	Gua	Cys	Met	Leu	-	Ala	Ala	-	hArg	Pro	Cys	Trp	63	26
GPS11380	Gua	Cys	Met	Leu	-	Arg	Val	-	5-Ava	Pro	Cys	Trp	>100	29
GPS11381	Gua	Cys	Met	Leu	-	Arg	Val	-	Abu	Pro	Cys	Trp	100	28

The bridge is between the yellow-colored amino acids. (*). Data were from 2 × 2 independent duplicates. If the variation between the results, altogether, was superior to 15%, the experiments were redone. None of the results are outside of those limits. Our internal data base for MCH-R1 binding experiments comprises 550 independent binding experiments over the course of a decade. For each plate of assay the internal control was MCH. (**) data of the parent compound. “**n’**” is the number of heavy atoms in the ring of the peptides comprising the bridge and the backbone of the amino acids between the colored ones. Abbreviations can be found in the legend of Appendix A.

**Table 2 molecules-26-01291-t002:** Binding affinities at human MCH-R1 of pseudopeptides derived from S38151, bearing a lactam bridge.

Code	Sequence												Km nM *	n’
	1	2	3	4	5	6	7	8	9	10	11	12		
S38151 (**)	Gua	Cys	Met	Leu	-	Arg	Val	Tyr	Arg	Pro	Cys	Trp	4,3	29
GPS13684	Gua	Glu	Met	Leu	-	Arg	Val	-	Arg	4-NH2-Pro	Orn	Trp	0,36	29
GPS13680	Gua	Glu	Met	Leu	-	Arg	Val	-	Me-Arg	Pro	Orn	Trp	0,4	29
GPS13673	Gua	Glu	Met	Leu	-	Arg	Val	-	Arg	Pro	Orn	Bta	0,68	29
GPS12744	Gua	Glu	Met	Leu	-	Arg	Val	-	Arg	Pro	Orn	Trp	0,75	29
GPS13663	Gua	Glu	Ethionine	Leu	-	Arg	Val	-	Arg	Pro	Orn	Trp	0,75	29
GPS13689	Gua	Glu	Met	Leu		Arg	Val		Arg	Pro	Orn	Tryptamide	0,87	29
GPS15288	Gua	Glu	Met	Leu	-	Arg	Val	Tyr	Arg	4-NH2-Pro	Orn	Trp	0,91	32
GPS13675	Gua	Glu	Met	Leu	-	Me-Arg	Val	-	Arg	Pro	Orn	Trp	0,95	29
GPS13682	Gua	Glu	Met	Leu	-	Arg	Val	-	hArg	Pro	Orn	Trp	1	29
GPS13683	Gua	Glu	Met	Leu	-	Arg	Val	-	Cav	Pro	Orn	Trp	1	29
GPS13665	Gua	Glu	Se-Met	Leu	-	Arg	Val	-	Arg	Pro	Orn	Trp	1,2	29
GPS14509	Gua	Glu	Se-Met	F3-Leu	-	Arg	Val	-	Arg	Pro	Orn	Trp	1,2	29
GPS13686	Gua	Glu	Met	Leu	-	Arg	Val	-	Arg	4-CF3-Pro	Orn	Trp	1,3	29
GPS13687	Gua	Glu	Met	Leu	-	Arg	Val	-	Arg	4-Ph-Pro	Orn	Trp	1,3	29
GPS13677	Gua	Glu	Met	Leu	-	hArg	Val	-	Arg	Pro	Orn	Trp	1,6	29
GPS14511	Gua	Glu	Met	Leu	-	Me-Arg	Val	-	Me-Arg	Pro	Orn	Trp	1,8	29
GPS13679	Gua	Glu	Met	Leu	-	Arg	Val	-	NO-Arg	Pro	Orn	Trp	1,9	29
GPS13667	Gua	Glu	Met	Nle	-	Arg	Val	-	Arg	Pro	Orn	Trp	2	29
GPS13674	Gua	Glu	Met	Leu	-	NO-Arg	Val	-	Arg	Pro	Orn	Trp	2	29
GPS13695	Gua	Glu	Met	Leu	-	Arg	Val	Tyr	Arg	Pro	Orn	Trp	2,2	32
GPS13666	Gua	Glu	Met	F3-Leu	-	Arg	Val	-	Arg	Pro	Orn	Trp	3	29
GPS13678	Gua	Glu	Met	Leu	-	Cav	Val	-	Arg	Pro	Orn	Trp	3	29
GPS13672	Gua	Glu	Met	Leu	-	Arg	Val	-	Arg	Pro	Orn	N-Me-Trp	3,4	29
GPS15287	Gua	Glu	Met	Leu	-	Arg	Val	-	Arg	Aib	Orn	Trp	3,4	29
GPS14514	Gua	Glu	Met	Leu	-	Arg	-	Tyr	Arg	Pro	Orn	Trp	3,5	29
GPS15292	Gua	Glu	Met	Leu	-	Arg	Val	Tyr	Arg	Pro	Orn	Trp	3,5	32
GPS13685	Gua	Glu	Met	Leu	-	Arg	Val	-	Arg	(Me)2-Pro	Orn	Trp	3,8	29
GPS13681	Gua	Glu	Met	Leu	-	Arg	Val	-	Cit	Pro	Orn	Trp	3,9	29
GPS14510	Gua	Glu	Met	F3-Leu	-	Me-Arg	Val	-	Arg	Pro	Orn	Trp	4,1	29
GPS12733	Gua	Glu	Met	Leu	-	Arg	Val	-	Arg	Pro	Dab	Trp	5,3	28
GPS14512	Gua	Glu	Met	Leu	-	Arg	Val	-	Me-Arg	Me3-Pro	Orn	Trp	5,7	29
GPS13676	Gua	Glu	Met	Leu	-	Cit	Val	-	Arg	Pro	Orn	Trp	6	29
GPS12739	Gua	Asp	Se-Met	Leu	-	Arg	Val	-	Arg	Pro	Orn	Trp	8,5	28
GPS14515	Aaba	Glu	Met	Leu	-	Arg	Val	-	Arg	Pro	Orn	Trp	8,6	29
GPS12743	Gua	Orn	Met	Leu	-	Arg	Val	-	Arg	Pro	Asp	Trp	9	28
GPS12746	Gua	Glu	Met	Leu	-	Arg	Val	-	Arg	Pro	Lys	Trp	12	30
GPS12745	Gua	Orn	Met	Leu	-	Arg	Val	-	Arg	Pro	Glu	Trp	13	29
GPS12737	Gua	Asp	Ethionine	Leu	-	Arg	Val	-	Arg	Pro	Orn	Trp	23	28
GPS13661	Gua	Glu	CH3-S-Cys	Leu	-	Arg	Val	-	Arg	Pro	Orn	Trp	26	29
GPS13668	Gua	Glu	Met	tBut-Gly	-	Arg	Val	-	Arg	Pro	Orn	Trp	28	29
GPS12731	Gua	Asp	Met	Leu	-	Arg	Val	-	Arg	Pro	Lys	Trp	41	29
GPS13664- Peak 1 (***)	Gua	Glu	Buthionine	Leu	-	Arg	Val	-	Arg	Pro	Orn	Trp	87	29
GPS13670	Gua	Ala	Met	Leu	-	Arg	Val	-	Arg	Pro	Ala	Trp	110	0
GPS12750	Gua	Asp	Met	Nle	-	Arg	Val	-	Arg	Pro	Orn	Trp	140	28
GPS13688	Gua	Glu	Met	Leu	-	Arg	Val	-	Arg	4-NH2-Pro	Ala	Trp	170	24
GPS12732	Gua	Glu	Met	Leu	-	Arg	Val	-	Arg	Pro	Dap	Trp	210	27
GPS13669	Gua	Glu	Met	Leu	-	Arg	Val	-	Arg	Pro	Orn	Trp	220	0
GPS13662	Gua	Glu	β -hMet	Leu	-	Arg	Val	-	Arg	Pro	Orn	Trp	250	30
GPS14488	Gua	Glu	Met	Leu	Gly	Arg	Val	-	Arg	Pro	Orn	Trp	270	32
GPS13664-Peak 2 (***)	Gua	Glu	Buthionine	Leu	-	Arg	Val	-	Arg	Pro	Orn	Trp	330	29
GPS11398	Gua	Asp	Met	Leu	-	Arg	Val	Tyr	Arg	Pro	Dap	Trp	380	29
GPS15293	arg	Glu	Met	Leu	-	Arg	Val	-	Arg	Pro	Orn	Trp	610	29
GPS11401	Gua	Asp	Met	Leu	-	Arg	Val	-	Arg	Pro	Dab	Trp	880	27
GPS14489	Gua	Glu	Met	Leu	-	Arg	Val	-	Cav	Pro	Orn	Trp	900	0
GPS12734	Gua	Asp	Met	Leu	Gly	Arg	Val	-	Arg	Pro	Orn	Trp	~10,000	31
GPS12736	Gua	Asp	β-hMet	Leu	-	Arg	Val	-	Arg	Pro	Orn	Trp	~10,000	29
GPS12742	Arg-BZ	Asp	Met	Leu	-	Arg	Val	-	Arg	Pro	Orn	Trp	~10,000	28
GPS13628	Gua	Asp	Met	4-hydro-Leu	-	Arg	Val	-	Arg	Pro	Orn	Trp	~10,000	28
GPS12752	Gua	Asp	Met	Leu	-	Arg	Val	-	Arg	Aib	Orn	Trp	~10,000	28
GPS12749	Gua	Asp	Met	F3-Leu	-	Arg	Val	-	Arg	Pro	Orn	Trp	~1000	28
GPS12751	Gua	Asp	Met	tBut-Gly	-	Arg	Val	-	Arg	Pro	Orn	Trp	~1000	28
GPS12735	Gua	Asp	CH3-S-Cys	Leu	-	Arg	Val	-	Arg	Pro	Orn	Trp	~1000	28
GPS12738	Gua	Asp	Buthionine	Leu	-	Arg	Val	-	Arg	Pro	Orn	Trp	~10,000	28
GPS11400	Gua	Asp	Met	Leu	-	Arg	Val	-	Arg	Pro	Dap	Trp	>10,000	26
GPS11408	Gua	Asp	Met	Leu	-	-	Ala	Ala	hArg	Pro	Dap	Trp	>10,000	26
GPS11410	Gua	Asp	Met	Leu	-	Arg	Val	-	Abu	Pro	Dap	Trp	>10,000	28
GPS14522	Gua	Glu	Met	Leu	-	Arg							>10,000	0
GPS11403	Gua	Asp	Met	Leu	-	Arg	Val	Tyr	hArg	Pro	Dap	Trp	>1000	29
GPS11399	Gua	Asp	Met	Leu	-	Arg	-	Tyr	Arg	Pro	Dap	Trp	>10,000	26
GPS11402	Gua	Asp	Met	Leu	-	Arg	Val	-	Arg	Pro	Orn	Trp	>10,000	28
GPS11404	Gua	Asp	Met	Leu	-	Arg	Val	Tyr	arg	Pro	Dap	Trp	>10,000	29
GPS11405	Gua	Asp	Met	Leu	-	Arg	Val	Ala	hArg	Pro	Dap	Trp	>10,000	29
GPS11406	Gua	Asp	Met	Leu	-	-	Val	Tyr	Arg	Pro	Dap	Trp	>10,000	26
GPS14523							Val	-	Arg	Pro	Orn	Trp	>10,000	0

The bridge is between the green-colored amino acids. When those cells are white, this denoted the absence of cycle for the control linear peptides: GPS13670, GPS13669, GPS14489, GPS14522 and GPS14523. Abbreviations can be found in the legend of Appendix A. **“n’**” is the number of heavy atoms in the ring of the peptides comprising the bridge and the backbone of the amino acids between the colored ones. (*) Data were from 2 × 2 independent duplicates. If the variation between the results, altogether, was superior to 15%, the experiments were redone. None of the results are outside of those limits. Our internal data base for MCH-R1 binding experiments comprises 550 independent binding experiments over the course of a decade. For each plate of assay the internal control was MCH. (**) data of the parent compound. (***) Because buthionine was not optically pure, the two subsequent pseudopeptides were separated, purified, and tested as “peak 1 and peak 2”.

**Table 3 molecules-26-01291-t003:** Binding affinities at human MCH-R1 of pseudopeptides derived from S38151, bearing a triazole bridge.

Code	Sequence												Km nM *	n’
	**1**	**2**	**3**	**4**	**5**	**6**	**7**	**8**	**9**	**10**	11	12		
S38151 (**)	Gua	Cys	Met	Leu	-	Arg	Val	Tyr	Arg	Pro	Cys	Trp	4,3	29
GPS14517	Gua	N3-hAla	Met	Leu	-	Arg	Val	-	Arg	Pro	Propargyl-Gly	Trp	2,1	29
GPS15290	Gua	N3-hAla	Met	Leu	-	Arg	bhVal	-	Arg	Pro	Propargyl-Gly	Trp	0,86	29
GPS15363	Gua	N3-hAla	Met	Leu	-	Arg	bhVal	-	Arg	Pro	Propargyl-Gly	Bta	0,3	29
GPS15364	Gua	N3-hAla	Met	Leu	-	Arg	bhVal	-	Arg	4-NH2-Pro	Propargyl-Gly	Trp	0,24	29
GPS15365	Gua	N3-hAla	SeMet	Leu	-	Arg	bhVal	-	Arg	Pro	Propargyl-Gly	Trp	0,34	29
GPS15366	Gua	N3-hAla	Met	Leu	-	Arg	bhVal	-	Arg	Pro	Propargyl-Gly	Triptamide	0,21	29
GPS15367	Gua	N3-hAla	NMeMet	Leu	-	Arg	bhVal	-	Arg	Pro	Propargyl-Gly	Trp	670	29
GPS15368	Gua	N3-hAla	Met	NMeLeu	-	Arg	bhVal	-	Arg	Pro	Propargyl-Gly	Trp	360	29
GPS18169	Gua	N3-hAla	Nle	Leu	-	Arg	bhVal	-	Arg	Pro	Propargyl-Gly	Trp	1,2	29
GPS14519	Gua	N3-nVal	Met	Leu	-	Arg	Val	-	Arg	Pro	Propargyl-Gly	Trp	2,3	29
GPS14516	Gua	Propargyl-Gly	Met	Leu	-	Arg	Val	-	Arg	Pro	N3-hAla	Trp	1,9	29
GPS14518	Gua	Propargyl-Gly	Met	Leu	-	Arg	Val	-	Arg	Pro	N3-nVal	Trp	1,8	29
GPS15291	Gua	Propargyl-Gly	Ethionine	Leu	-	Arg	bhVal	-	Arg	Pro	N3-hAla	Trp	91	29

The bridge is between the blue-colored amino acids. Abbreviations can be found in the legend of Appendix A. **“n’**” is the number of heavy atoms in the ring of the peptides comprising the bridge and the backbone of the amino acids between the colored ones. The red cells correspond to the most active compounds chosen for in vivo studies. (*) Data were from 2 × 2 independent duplicates. If the variation between the results, altogether, was superior to 15%, the experiments were redone. None of the results are outside those limits. Our internal data base for MCH-R1 binding experiments comprises 550 independent binding experiments over the course of a decade. For each plate of assay the internal control was MCH. (**) data of the parent compound.

**Table 4 molecules-26-01291-t004:** Functionality of the most active antagonists pseudopeptides at MCH-R1.

Code	Sequence												Km (nM)	n’	Ki (pM)
S38151	Gua	Cys	Met	Leu	-	Arg	Val	Tyr	Arg	Pro	Cys	Trp	4.3	29	5000
GPS14517	Gua	N_3_-hAla	Met	Leu	-	Arg	Val	-	Arg	Pro	hPra	Trp	2.1	29	22
GPS14518	Gua	hPra	Met	Leu	-	Arg	Val	-	Arg	Pro	N_3_-nVal	Trp	1.8	29	27
GPS15290	Gua	N_3_-hAla	Met	Leu	-	Arg	βhVal	-	Arg	Pro	hPra	Trp	0.86	29	28
GPS18169	Gua	N_3_-hAla	Nle	Leu	-	Arg	βhVal	-	Arg	Pro	hPra	Trp	1.2	29	28
GPS13695	Gua	Glu	Met	Leu	-	Arg	Val	Tyr	Arg	Pro	Orn	Trp	2.2	32	130
GPS13683	Gua	Glu	Met	Leu	-	Arg	Val	-	Canava	Pro	Orn	Trp	1	29	150
GPS14519	Gua	N_3_-nVal	Met	Leu	-	Arg	Val	-	Arg	Pro	hPra	Trp	2.3	29	430
GPS13671	Gua	hPra	Ethionine	Leu	-	Arg	Val	-	Arg	Pro	N_3_-hAla	Trp	120	28	8000
GPS13670	Gua	Ala	Met	Leu	-	Arg	Val	-	Arg	Pro	Ala	Trp	110	0	30,000
GPS13669	Gua	Glu	Met	Leu	-	Arg	Val	-	Arg	Pro	Orn	Trp	220	0	300,000

The bridge is between the colored amino acids: yellow for an S-S bridge, blue for a triazole bridge, green for a lactam bridge. The white cells correspond to linear control peptides. S38151 (in red letters) is the internal reference peptide. **“n’**” is the number of heavy atoms in the ring of the peptides comprising the bridge and the backbone of the amino acids between the colored ones. Our internal data base for MCH-R1 antagonism measurement comprises 300 independent experiments over the course of a decade. Abbreviations can be found in the legend of Appendix A.

**Table 5 molecules-26-01291-t005:** Plasma stability (t_1/2_, half-life) of some of the peptides in the series (mouse, rat, human).

	S38151	GPS12744	GPS13663	GPS13684	GPS15290	GPS18169
Mouse	23 ± 1.8	26 ± 3	19 ± 1	26 ± 1	173 ± 9	175 ± 17
Rat	60 ± 3	143 ± 12	94 ± 3	135 ± 9	300 *	300 *
Human	14 ± 1	300 *	300 *	300 *	300 *	300 *

The results are expressed in minutes and are the mean of at least 3 independent experiments. * 300 (5 h) is the maximal time of the experiments. This means that under those conditions, the pseudopeptides were not metabolized at all, and 100% of the initial dose was found. S38151 is the reference peptide bearing an S-S bridge; GPS12744, GPS13663 and GPS13684 possess a lactam bridge and various substitutions in the sequence; GPS15290 possesses a triazole bridge (see Appendix A for peptide sequences). Because the experiment on GPS18169 was obtained in a separate set of experiments, the results were presented apart from the main table.

**Table 6 molecules-26-01291-t006:** Blood biochemistry of diet-induced obese mice with and without treatment by the MCH-R1 antagonist GPS18169.

			Insulin			Total Cholesterol			Triglyceride		
			pg/mL			mg/dL			mg/dL		
			Day 56	Day 113	Day 141	Day 56	Day 113	Day 141	Day 56	Day 113	Day 141
Vehicle (Normal Diet)	(*)	Mean	381.1	438.3	760.7	98.1	103.1	101.2	69.5	86.3	78.3
		SEM	51.7	57.2	79.7	2.4	2.8	4.5	4.8	5.3	6.4
Vehicle (High-Fat Diet)	(*)	Mean	887.0	2543.0 †	2016.6 †	155.6 †	163.6 †	157.7 †	123.4 †	122.1 †	123.5 †
		SEM	126.9	383.3	477.2	8.9	8.9	17.5	6.7	5.7	11.6
GPS18169 (High-Fat Diet)	(*)	Mean	1068.0	1549.7	957.0 *	155.6	135.7	132.1	122.9	108.3	102.1
		SEM	145.9	370.8	371.8	7.4	9.3	6.6	3.5	6.0	3.3
GPS18169 (High-Fat Diet)	(**)	Mean	909.5	1983.8	760.4 *	158.3	143.1	135.6	121.8	115.3	108.9
		SEM	125.6	554.7	201.4	7.0	13.9	14.6	4.6	6.0	9.4

(*) 10 mg/kg daily (QD) x 12 weeks; (**) 5 mg/kg QD x 12 weeks; Two-way ANOVA followed by Bonferroni test was applied for comparison between the vehicle and treated groups at each time point. Differences are considered significant at ^†^
*p* < 0.05, vs. Vehicle (Normal Diet); * *p* < 0.05, vs. Vehicle (High-fat Diet) (see red cells). Complete individual data can be found in Appendix A (Blood chemistry). SEM is standard error of the mean.

**Table 7 molecules-26-01291-t007:** Comparison of the adipose tissue weights between lean, high fat diet fed and GPS18169-treated mice.

Treatment	Dose		BW (g) Day 141	Adipose Tissue Weight (g)									
				Epididymal		Mesenteric		Retroperitoneal		Inguinal		Brown Fat	
			g	g	% BW	g	% BW	g	% BW	g	% BW	g	% BW
Vehicle (Normal Diet)	10 mL/kg QD x 12 wks	Mean	27.2	0.735	2.66	0.345	1.26	0.219	0.79	0.402	1.46	0.099	0.36
		SEM	0.5	0.063	0.20	0.021	0.06	0.025	0.08	0.034	0.11	0.006	0.02
Vehicle (High-fat Diet)	10 mL/kg QD x 12 wks	Mean	40.0 †	1.972 †	4.98 †	0.688 †	1.68	0.779 †	1.87 †	1.725 †	4.16 †	0.164 †	0.40
		SEM	2.6	0.152	0.30	0.082	0.11	0.116	0.17	0.222	0.33	0.018	0.02
GPS18169-002	10 mg/kg QD x 12 wks	Mean	29.9 *	1.991	6.71	0.334 *	1.08 *	0.320 *	1.01 *	0.892 *	2.77	0.084 *	0.27 *
		SEM	1.4	0.141	0.47	0.047	0.11	0.064	0.17	0.206	0.55	0.009	0.02
GPS18169-002	5 mg/kg QD x 12 wks	Mean	29.8 *	1,882	6.67	0.351 *	1.08 *	0.321 *	0.97 *	0.922 *	2.69	0.096 *	0.30 *
		SEM	2.4	0.148	0.66	0.094	0.19	0.093	0.21	0.314	0.73	0.023	0.04

One-way ANOVA followed by Dunnett’s test was applied for comparison between the vehicle and treated groups at each time point. Differences are considered significant at †*p* < 0.05, vs. Vehicle (Normal Diet); * *p* < 0.05, vs. Vehicle (High-fat Diet). In red characters are all the data significantly different from their corresponding control experiments. Complete individual data can be found in Appendix A.

**Table 8 molecules-26-01291-t008:** The study design is summarized in the following study design summary table.

Group	Test	Route	Conc.	Dosage		Mice
	Article		mg/ml	mL/kg	mg/kg	(Male)
1	Vehicle ^a,b^ (Normal Diet)	IP	NA	10	NA, QD x 12 weeks	10
2	Vehicle ^a,c^ (High-fat diet)	IP	NA	10	NA, QD x 12 weeks	10
3	GPS18169 ^a,c^	IP	1	10	10, QD x 12 weeks	10
4	GPS18169 ^a,c^	IP	0.5	10	5, QD x 12 weeks	10

^a^ Water for injection. ^b^ Group 1 mice are placed in a normal diet (MFG) for 8 weeks before treatment. ^c^ Groups 2–4 mice are placed in a high-fat diet (HFD, 60% of calories) for 8 weeks before treatment. Test articles and vehicle are administered daily by intraperitoneal injection (IP) for 12 weeks after 8 weeks of HFD feeding.

## Data Availability

All the individual data are gathered in the Appendix A section.

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
