# Peer review of "MCH-R1 Antagonist GPS18169, a Pseudopeptide, Is a Peripheral Anti-Obesity Agent in Mice"

_molecules, 2021, doi:10.3390/molecules26051291_

Round 1
Reviewer 1 Report
In this study, the authors described a formidable series of experiments to test the effectiveness of various MCH-R1 antagonists, as potential therapeutics for obesity. One such compound, GPS18169, shows activity at the MCH-R1, and is effective at reducing weight gain in mice fed a high fat diet. These are, undoubtedly, important observations, but in the current format, the manuscript is incredibly difficult to read and follow. The authors need to extensively edit the manuscript, and I would suggest using and English editing service, as there are numerous examples of unconventional language.
Some specific comments:
1) The authors might consider placing the large tables of binding affinities as supplemental data, to reduce the length of the manuscript.
2) The abstract needs to be re-written. At present, it is more like review article précis, and needs to be more focussed. In addition, the Introduction needs to be far more precise - the language is overly flamboyant, which makes it very difficult to follow.
3) The Results section requires a lot of editing. The experimental sections using both cell line and mouse work, do not describe the experiments performed with sufficient clarity. It is unclear, sometimes, when the authors are describing current data, or their previous work ("our past record of these assays").
4) Figure 3 requires a much better figure legend. There is no description of the experiments, or the data presentation or analysis. Are these means+/- SEM, SD? Where are the SEM/SD on the graphs? I suggest separating out the results from the two compounds investigated in to separate figures - or if not, to make the labelling much clearer.
5) Figure 4 and 5 - the graphs need increasing in size.
6) The discussion requires more critical interpretation, and focus.
Reviewer 2 Report
Please see attached Word document.

Round 2
Reviewer 1 Report
The presentation and readability of this revised manuscript is much improved, making it far easier to read. I appreciate the authors' comments with regards to formatting issues. In addition, I agree that there is so much data in this study that the structure activity relationships could have been a separate manuscript. Thank you for the efforts you have made in improving the clarity.